

# Pollen-based reconstruction of spatially-explicit vegetation cover over the Tibetan Plateau since the last deglaciation

Pengchao Zhang[1,2], Yi Luo[2,3], Dan Liu[2], Xiaoyi Wang[2], Tao Wang[2*]

[1]College of Ecology, Lanzhou University, Lanzhou 730000, China
[2]State Key Laboratory of Tibetan Plateau Earth System, Environment and Resources (TPESER), Institute of Tibetan Plateau Research, Chinese Academy of Sciences, Beijing 100101, China
[3]University of Chinese Academy of Sciences, Beijing 100049, China.

*Correspondence to*: Tao Wang (twang@itpcas.ac.cn)

**Abstract.** Spatiotemporally contiguous paleo-vegetation reconstructions are essential for studying climate-vegetation interactions, providing critical data for paleoclimate modeling, and refining past land cover in Earth System Models (ESMs) and scenarios of anthropogenic land-cover changes (ALCCs). Here, we present the first spatiotemporally contiguous paleo-vegetation cover dataset for the Tibetan Plateau, spanning from the last deglaciation (16 ka) to the preindustrial era. This dataset was achieved using two sets of random forest (RF) models: one focused on temporal reconstructions (RF-temporal)

and the other on spatial reconstructions (RF-spatial). RF-temporal reconstructs temporal trends from 61 fossil pollen records across the Tibetan Plateau, while RF-spatial interpolates site-based cover, producing a dataset with a spatial resolution of 0.5° × 0.5° and a temporal resolution of 400 years. The dataset provides estimates of vegetation cover, along with standard errors, for three vegetation types (vegetation, woody plant, and herbaceous plant). To illustrate, we present the temporal trends and spatial distribution of vegetation cover for these vegetation types, comparing them with the vegetation cover used in ESMs.

We further discuss the dataset's reliability and applications, along with the discrepancies between our reconstructed results and those used in ESMs, highlighting possible reasons for these differences.

## 1 Introduction

Climate affects vegetation distribution and structure, while vegetation, in turn, influences climate through biogeophysical effects, including changing albedo (Alibakhshi et al., 2020), roughness (Thomas and Foken, 2007), and evapotranspiration

(Yan et al., 2012), and biogeochemical effects, including changing greenhouse gases (CH4 and CO2) (Gui et al., 2024). Therefore, the spatiotemporal dynamics of vegetation cover serve as crucial boundary conditions driving global climate models (GCMs) and Earth system models (ESMs). Reconstructing the past spatiotemporal dynamics of vegetation cover not only aids in understanding the responses and feedbacks of vegetation to climate change but also provides foundational data for these models and anthropogenic land-cover changes (ALCCs) (e.g., KK10 and HYDE) (Githumbi et al., 2022; Li et al.,





2023). Long-term vegetation cover data can be derived from paleo-vegetation records in stratigraphic sediments (e.g., fossil pollen) reconstructions and dynamic global vegetation models (DGVMs) simulations.

Fossil pollen, as a direct proxy for past vegetation, has been widely used to reconstruct paleo-vegetation cover. Early methods for reconstructing vegetation changes using qualitative (Biomization) (Sun et al., 2020) and semi-quantitative methods (relative changes in different biomes) (Zhao et al., 2017). Subsequently, researchers employed the Landscape

Reconstruction Algorithm (LRA), which corrects for the non-linear relationship between pollen abundance and vegetation cover, such as through the "Regional Estimates of VEgetation Abundance from Large Sites" (REVEALS) (Sugita, 2007), to quantitatively reconstruct vegetation cover changes. While, REVEALS was mainly developed to estimate vegetation cover changes from fossil pollen deposited in large lakes (>50 ha), and from multiple small-sized sites (Marquer et al., 2017; Githumbi et al., 2022; Li et al., 2023). However, it is still challenging to obtain a spatio-temporally explicit estimate of

vegetation cover changes. The outputs from REVEALS represent the proportion of different vegetation types within the vegetation area, and it still requires correction using DGVMs' estimates of total vegetation cover or bare ground cover to obtain the actual cover of vegetation types (Strandberg et al., 2023). Although these outputs are useful for summarizing paleo-vegetation changes over time based on pollen assemblages, they are of limited utility when spatially continuous data or actual vegetation cover is required.

The ESMs or DGVMs use mathematical representations of the physical, chemical, and biological principles to simulate how vegetation varies with climate and $CO_2$ concentration (Braghiere et al., 2023; Chen et al., 2023). However, these models often do not activate dynamic vegetation processes but use prescribed vegetation cover. For example, in PMIP4 simulations, only one model activated vegetation dynamics, while other models used prescribed preindustrial vegetation cover, due to the lack of a comprehensive and reliable vegetation dataset during these paleo periods (Jungclaus et al., 2017; Kageyama et al.,

2018). ESMs with vegetation dynamics could simulate potential vegetation distributions corresponding to paleoclimate, but the model outputs are often fraught with notorious uncertainties in paleoclimate variables (Brierley et al., 2020). Machine learning approaches such as the modern analogy technique (MAT) (Davis et al., 2024) have been increasingly used to reconstruct past vegetation dynamics from fossil pollen records at the biome level (e.g., Sobol et al., 2019; Lindgren et al., 2021). These machine learning methods (e.g., random forest, extreme gradient boosting, and k-nearest neighbor) do not

require prior knowledge, can quickly learn relationships within data, and are adept at handling nonlinear relationships and high-dimensional data (Sobol et al., 2019b; Lindgren et al., 2021).

The Tibetan Plateau is of particular interest as a global region where the westerlies and Asian monsoons converge, making it a climate-sensitive area with noticeable vegetation responses to climate change (Wang et al., 2021). Additionally, due to its unique geographical position, the plateau's terrestrial ecosystem plays a crucial role as an ecological security barrier (Chen et

al., 2021; Wang et al., 2024). Even small changes in vegetation can have significant effects on local and broader Asian climates, potentially influencing other global climate-sensitive regions, such as the Arctic, through teleconnections (Tang et al., 2023a, 2024). Understanding the response and feedback of plateau paleo-vegetation to climate change from the last



Glaciation to the present can provide essential insights into potential vegetation changes under future climate scenarios (Zhao et al., 2015; McElwain, 2018; Nolan et al., 2018).

Here we reconstructed spatiotemporally contiguous vegetation cover changes at a regional scale using machine learning algorithms. Specifically, we first used a temporally random-forest model (RF-temporal) to reconstruct the cover of different vegetation types from fossil pollens at the site level. We then employed a spatially RF model (RF-temporal) to obtain a spatially contiguous dataset. The generated dataset provided vegetation cover data for the Tibetan Plateau from the Last deglaciation (16 ka BP) to the present, with a temporal resolution of 400 years and a spatial resolution of 0.5°, covering

different vegetation types (including vegetation, woody, and herbaceous). This dataset will be expected to enhance our understanding of paleovegetation dynamics and its response to climate change on the Tibetan plateau. More importantly, this dataset could provide the vegetation boundary condition for ESMs that are used to simulate paleoclimate changes and resultant biogeochemical and biophysical impacts.

## 2. Data and methodology

### 2.1 Fossil and modern pollen datasets

**Fossil pollen datasets**: The fossil pollen dataset was obtained from Cao et al. (2022), including 65 records and 4395 samples with 143 harmonized pollen taxa. The age-depth model for each pollen record was reconstructed using Bayesian age-depth modeling and the IntCal09 radiocarbon calibration curve (detailed information about the standardized chronology is presented in Cao et al. (2013)). This dataset was selected using the following criteria to ensure data quality and an

adequate site distribution: (1) each record had more than three chronological controls; (2) the duration of the record was more than 2,000 years; (3) the sampling resolution was finer than 1,000 years. We followed the harmonized taxonomy table published by Herzschuh et al. (2022) to harmonize 245 pollen taxa into 125 taxa. The selected records are evenly distributed across the Tibetan Plateau (Fig. 1), and the number of samples increased from 20 during the deglaciation to 100 in the Late Holocene, meeting the requirements for reconstructing the past spatiotemporal patterns of vegetation on the Tibetan Plateau.

**Modern pollen datasets**: To encompass as many scenarios as possible of different vegetation combinations under varying climate conditions within the stratigraphic fossil record (Wang et al., 2023), the modern pollen datasets were obtained from the 'Modern pollen dataset for Asia' (Cao et al., 2022). This dataset covers eastern and northern Asia including 9,772 sampling sites across eastern and northern Asia, covering 242 harmonized pollen taxa, which represent the diverse vegetation types across most of Asia (Fig. S1). After harmonizing the dataset, we selected taxa shared with the fossil pollen

dataset, resulting in 125 taxa and 9,769 in sample, which were then proportionally standardized to 100%.

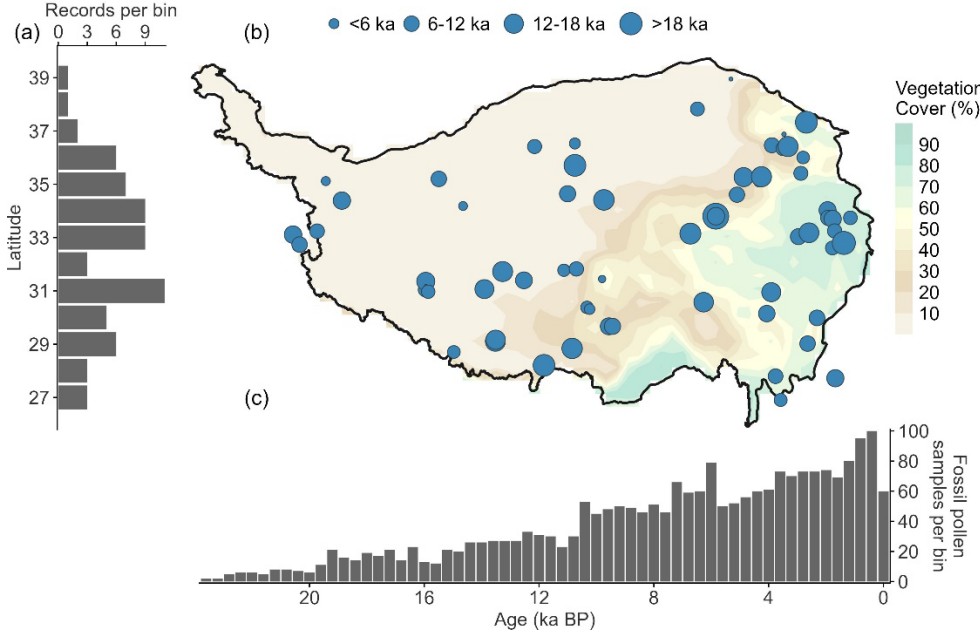

**Figure 1: Locations and temporal coverage of the fossil pollen records.** (a) Latitudinal distribution of fossil pollen records, as well as their (b) site locations. (c). Temporal coverage of the fossil pollen samples, binned at 400-year intervals. Years before the present (ka BP) are relative to 1950 CE. The bubble diameter corresponds to the temporal coverage of each record.

## 2.2. Vegetation cover data

**Modern vegetation cover**: Modern vegetation cover data were obtained from the Global Land Surface Satellite (GLASS) fractional vegetation cover products (http://www.glass.umd.edu/FVC/MODIS/500m/) (Jia et al., 2015). This dataset provides global vegetation cover data at a 500m pixel resolution, with extensive validation from high-resolution satellite data and ground measurements demonstrating high accuracy (Liu et al., 2019). In this study, the average annual maximum vegetation cover from 2000 to 2020 was used to represent modern vegetation cover. For the 9,092 modern topsoil pollen samples, circular buffers were applied, with a 5 km radius for surface soil samples and a 50 km radius for lake surface sediments. The average vegetation cover within each buffer was used to represent the cover associated with each pollen record. Since the modern pollen dataset does not distinguish land use at the site level, the Global Lakes and Wetlands Database: Lakes and Wetlands Grid (Level 3) (Lehner and Döll, 2004) was used to identify pollen originating from lake sources. Among the samples, 1,192 pollen records were from lakes and 8,577 were from topsoil. To further differentiate vegetation cover by type, we used the MODIS Land Cover Type Product (MCD12Q1), which provides an annual Plant Functional Type (PFT) classification (DiMiceli et al., 2022). Trees were classified as "woody", while shrubs and grasses were grouped as "herbaceous." The proportion of each vegetation type's area within the circular buffer of each modern topsoil pollen sample





relative to the total vegetation area was calculated, and this proportion, multiplied by the total vegetation cover, provided the cover of each specific vegetation type.

**Paleo-vegetation cover**: Paleo-vegetation cover derived from seven ESMs from CMIP4 project (Kageyama et al., 2018), TraCE-21k-II (He and Clark, 2022), and Hopcroft 与 Valdes (2021), including models ACCESS-ESM1.5, CESM2, INM-CM4.8, IPSL-CM6A-LR, MPI-ESM1.2-LR, TraCE-21K-II (CCSM3), HadCM3B (Table 1) are evaluated in this study. Among these, TraCE-21K-II (CCSM3) and HadCM3B are transient simulations with dynamic vegetation, while MPI-ESM1.2-LR is a snapshot simulation with dynamic vegetation. The remaining four models run with prescribed preindustrial vegetation cover due to the lack of a comprehensive and reliable global vegetation dataset (Otto-Bliesner et al., 2017). For comparison with the reconstruction results in this study, we standardized all forest vegetation types in the models as woody and all grass and shrub vegetation types as herbaceous.

**Table 1** Earth system models used in this study.

| Institution | Model name | Spatial resolution | Vegetation cover |
| --- | --- | --- | --- |
| CSIRO | ACCESS-ESM1.5 | 1.875°×1.25° | Prescribed* |
| NCAR | CESM2 | 1.25°×0.9375° | Prescribed* |
| INM | INM-CM4.8 | 2°×1.5° | Prescribed* |
| IPSL | IPSL-CM6A-LR | 2.5°×1.6° | Prescribed* |
| MPI | MPI-ESM1.2-LR | 1.875°×1.875° | Diagnostic |
| NCAR | TraCE-21K-II (CCSM3) | 3.75°×3.75° | Diagnostic |
| MOHC | HadCM3B | 3.75°×2.5° | Diagnostic |

## 2.3. Paleoclimate data

Paleoclimate data were taken from the CHELSA TraCE21k database (Karger et al., 2023), which offers high spatial (30 arc seconds) and temporal (centennial time slices) resolution. CHELSA TraCE21k uses a similar algorithm to CHELSA (Climatologies at High Resolution for Earth's Land Surface Areas) to process TraCE21k data, and it has been corrected using modern data. In this study, we resample the spatial resolution to 0.5° × 0.5° using bilinear interpolation and averaged the temporal resolution to 400 years, ensuring consistency in both spatial and temporal scales with pollen records.

## 2.4. Random Forest

Random forest, as an advanced machine learning technique, is known for its high accuracy and efficiency in handling high-dimensional data, making it widely used in the field of ecology (Wang et al., 2023; Liu et al., 2024). In this study, we employed two sets of random forest models. The first set of RF models (RF-temporal) was used to reconstruct the temporal trends of vegetation cover corresponding to fossil pollen. Based on these results, the second set of RF models (RF-spatial)



predicted the spatial distribution of vegetation cover across the Tibetan Plateau by analyzing the spatial relationships between point-based vegetation cover, climate, and terrain data (Fig. 2).

Specifically, in the first model, we used modern pollen percentages and terrain variables as predictors (Table S1), and the cover of different vegetation types as the response variable to predict the vegetation cover represented by fossil pollen at

different periods. During the model-building process, we selected the model with the lowest error as the optimal model based on the coefficient of determination ($R^2$) and root mean square error (RMSE) through ten-fold cross-validation. To ensure the robustness of the prediction results, the optimal model was run 500 times with varying random seeds, and the average of these 500 runs was taken as the final vegetation cover estimate.

For the second set of models, RF-spatial models were constructed at 400-year bins from the last deglaciation to the present

for each vegetation type. The predictors included 55 climate variables and 8 terrain variables (Table S2), while the response variable was the vegetation cover at fossil pollen sites. Similar to the RF-temporal, after determining the optimal model, we ran the model 500 times to obtain the uncertainty and confidence intervals of the spatial predictions. To address potential errors inherent in the RF model itself, we applied the classical Delta Method to perform bias correction using modern vegetation cover data (Beyer et al., 2020; Karger et al., 2023). Additionally, to account for extreme scenarios caused by

uncertainty propagation, we used the 5th and 95th percentiles of vegetation cover at each fossil pollen site in RF-temporal as response variables in the RF-spatial, yielding a conservative estimate of uncertainty.

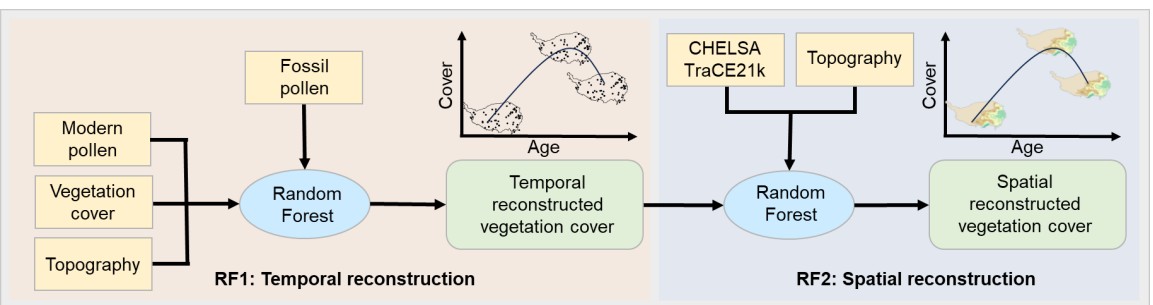

**Figure 2: Summary of major steps used in reconstructing vegetation cover using the random forest.**

## 3 Results

### 3.1 Paleochanges in vegetation cover over the Tibetan Plateau

By comparing MAT and five ML algorithms, we found that the random forest (RF) algorithm performed the best, achieving the highest goodness of fit and the lowest error (Fig. S2). Consequently, this study selected RF to reconstruct the spatiotemporal changes in vegetation cover across various vegetation types on the Tibetan Plateau over the past 16,000 years (see Method). The ten-fold cross-validation showed that the RF model achieved a high accuracy in the reconstruction of

vegetation cover (Fig. 3). For temporal reconstruction of different vegetation types, the $R^2$ values for total vegetation, woody,

and herbaceous cover were 0.82, 0.84, and 0.69, respectively. In spatial reconstruction, the R² values for these types were 0.89, 0.75, and 0.63.

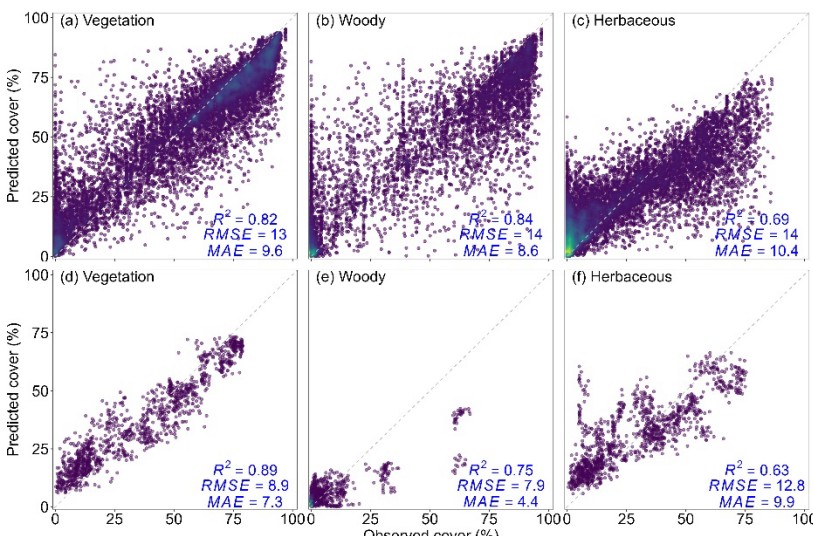

**Figure 3: Performance of the RF-temporal (a-c) and RF-spatial (d-e).** Each panel presents the relationship between predicted versus train vegetation cover values for each model based on 10-fold cross-validation. The dashed line represents the 1:1 line.

Based on the above-mentioned spatiotemporal model, we reconstructed the spatial-temporal changes in the different vegetation types at a temporal resolution of 400 years from the Last Deglaciation period (16 ka BP) to the present (see Methods). For total vegetation cover, the coverage over the past 16 ka BP varied by more than 8%, approximately half of the present-day cover (18% ± 0.5%) (Fig. 4a). Throughout the past 16,000 years, total vegetation cover reached its lowest value (14% ± 0.6%) during the Last Deglaciation period (15 ka BP), gradually peaking (22% ± 0.7%) during the warmest period of the Mid-Holocene (~8 ka BP), and then gradually declining toward the present-day (18% ± 0.5%).

Specifically, changes in vegetation cover reveal a distinct three-phase pattern that generally corresponds to climatic shifts. The first phase coincides with the Bølling-Allerød warm event (14.7–12.9 ka BP) when rapid warming of ~1.5℃ within a millennium led to a 5% rise in vegetation cover. This trend was interrupted by the Younger Dryas cold event (12.9–11.7 ka BP) when vegetation on the Tibetan Plateau was primarily concentrated in the southeastern region (Fig. S3). During the second phase, from the Early Holocene (11.7 ka BP) to the Mid-Holocene (8 ka BP), vegetation cover gradually increased, reaching its peak value (22% ± 0.7%), which is 2% higher than the present (18% ± 0.5%). At this time, vegetation on the Tibetan Plateau expanded further from the southeast to the western and northern regions (Fig. S3). Throughout the third phase, from the Mid-Holocene (8 ka BP) to the preindustrial era, the climate experienced a period of steady cooling with fluctuating warm and cold phases, resulting in a gradual decrease in vegetation cover. During this period, the spatial patterns

of vegetation cover on the Tibetan Plateau were similar to those of the modern, with slight differences, notably a decline in vegetation cover in the northern plateau (Fig. S3).

For woody cover, the variation over the entire period from 16 Ka BP to the present is 3%, approximately three-quarter of the
present-day cover (4% ± 0.2%) (Fig. 4b). The temporal changes in woody plant cover reveal a distinct two-phase pattern. The first phase spans from the Last Deglaciation period (16 ka BP) to the Early Holocene (9 ka BP), during which woody plant cover rapidly increased, reaching its peak value (6% ± 0.2%), which is 2% higher than the present (4% ± 0.2%). like vegetation cover, the response of woody cover to millennial-scale climate events (BA and YD) is pronounced. During this phase, forests expanded from the southernmost edge of the plateau to the southeastern margin. Throughout the second phase,
from the Early Holocene (11 ka BP) until the preindustrial era, woody cover experienced a steady decline, decreasing by approximately 2% over the entire period. Spatially, although forests remained distributed along the southeastern margin, the overall area of distribution contracted compared to the Mid-Holocene.

For herbaceous cover, the variation over the entire period is 7%, approximately half of the present-day cover (15% ± 0.2%) (Fig. 4c). The temporal changes in herbaceous cover exhibit a three-phase pattern similar to that of vegetation cover: a rapid
increase during the BA event, a steady rise from the Last Deglaciation period (13 ka BP) to the Mid-Holocene (6 ka BP), and a gradual decrease in the third phase (6-0 ka BP) that eventually stabilized. Spatially, during the Deglaciation period, herbaceous plants were primarily distributed in the southeastern part of the plateau. By the Mid-Holocene, their distribution expanded eastward and northward, followed by a retreat back towards the southeast in the present-day.

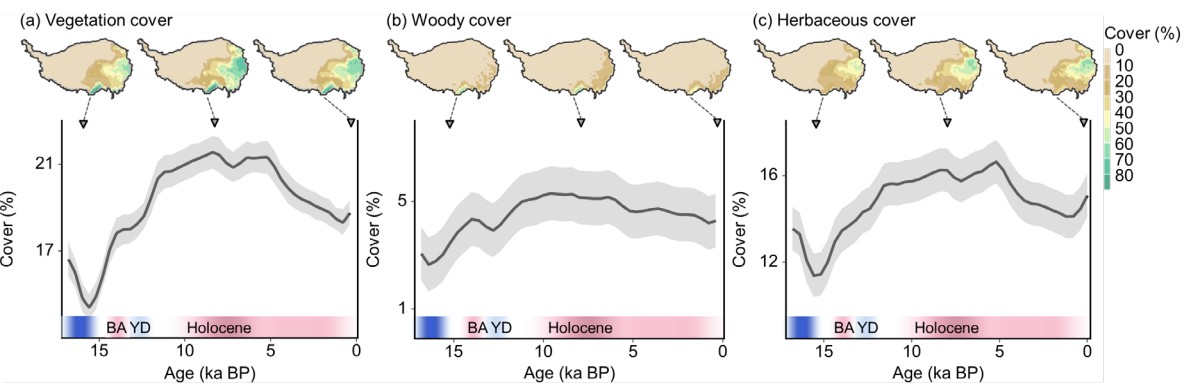

**Figure 4: Changes in vegetation cover on the Tibetan plateau since 16 Ka BP.** Total vegetation cover (a), Woody cover (b), Herbaceous cover (c). The solid lines represent the smoothed changes in woody cover at a 1000-year resolution. The shading indicates the confidence interval (5%-95%) obtained by 500 optimal RF models. Shown at the top are spatial distributions of vegetation cover at 15 Ka BP, 7.5 Ka BP, and 0 Ka BP.





## 3.2 The comparison of pollen-based reconstruction with model outputs

Compared to the reconstruction, most of the models have poor performance in capturing the spatial pattern of vegetation cover for the Mid-Holocene (6 ka BP). Only ACCESS-ESM1.5 and INM-CM4.8 generally capture this pattern (Fig. 5), with spatial correlations of 0.9 and 0.6, respectively. These two models have correctly simulated high vegetation cover in the southeastern TP and low vegetation cover in the northwestern TP. By contrast, other ESMs overestimate the spatial extent of vegetation cover, especially in the northwestern TP and southeastern TP. The spatial correlations were relatively low, ranging from 0.27 for CESM2 to 0.55 for MPI-ESM1.2-LR, partly because the models simulated vegetation cover in the northwestern TP where vegetation is absent according to pollen-based reconstruction.

In terms of vegetation types, most models can capture the spatial pattern of woody cover, with spatial correlations ranging from 0.49 (INM-CM4.8) to 0.72 (CESM2), with the woody cover mainly distributed along the southeastern edge of the plateau. By contrast, except for ACCESS-ESM1.5 and INM-CM4.8, there is a notorious bias in the simulation of herbaceous cover, with spatial correlation coefficient ranging from -0.56 (TraCE-21K-II) to 0.33 (MPI-ESM1.2-LR)). They failed to capture the pollen-based spatial pattern, with high cover in the east and low cover in the west. These models either simulated an opposite spatial pattern (e.g., TraCE-21K-II and HadCM3B), or a homogenized high cover across the entire plateau (e.g., CESM2 and IPSL-CM6A-LR). The model-data comparison suggested a general overestimation of the spatial extent of herbaceous cover, particularly in the western plateau. This model-data discrepancy primarily contributes to the total vegetation cover (Fig. S4).



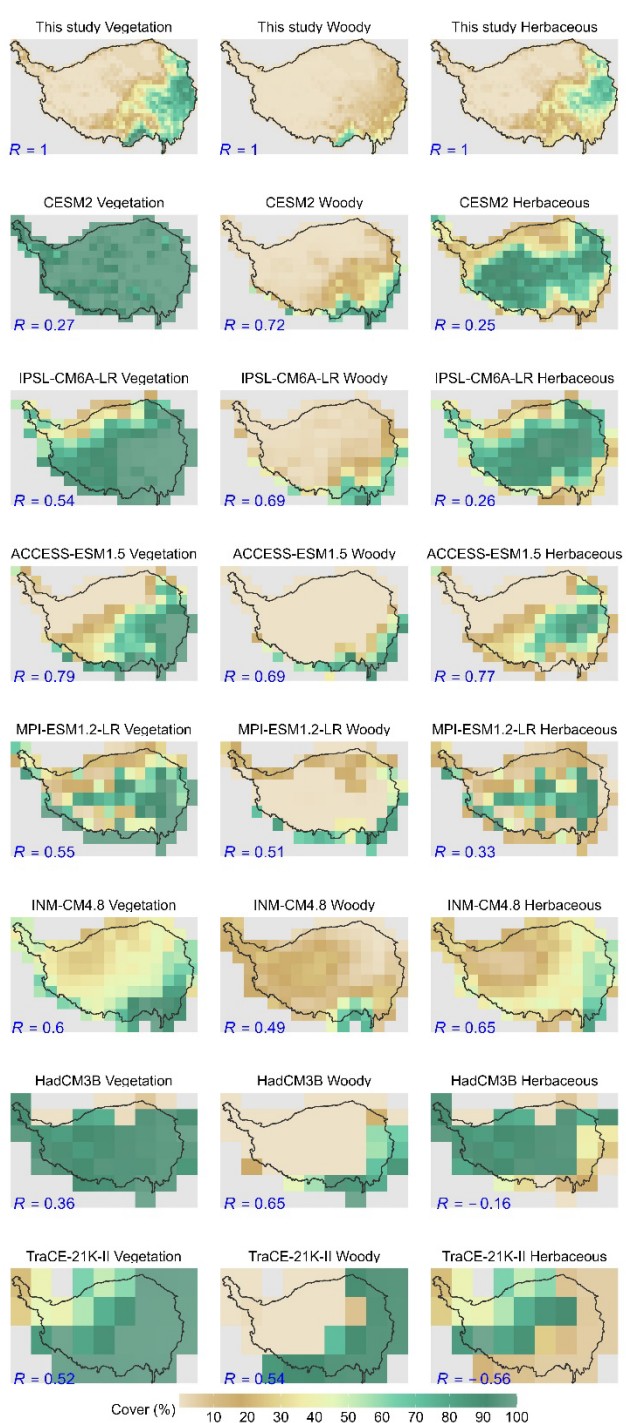

**Figure 5: Spatial distribution of vegetation cover from model-prescribed or simulations and the reconstructed dataset for the Mid-Holocene (6 Ka BP).** The numerical values in the lower left of each panel indicate the spatial correlation between the reconstructed data and model-prescribed or simulated cover.



In terms of variations at the centennial timescale, the pollen-based reconstruction shows an increase from 16 to 8 ka BP, followed by a decline from 8 to 0 ka BP. While the model simulations display differing temporal patterns. In HadCM3B, vegetation cover rises from 10 ka BP, reaching its peak at 6 ka BP, and then remains stable due to relatively steady woody

and herbaceous cover. In contrast, TraCE-21k-II largely captures a similar temporal trend with that from pollen reconstruction, but the decline from 8 ka BP to the present is primarily driven by a decrease in woody cover, whereas the reduction in total vegetation cover from pollen-based reconstruction is mainly due to a decrease in herbaceous cover. In PMIP4, the model-prescribed vegetation cover for the Mid-Holocene exhibits significant variability, with vegetation cover across different models ranging from 41.3% to 97%. This substantial difference in vegetation cover between models

primarily stems from herbaceous cover, which ranges from 25% to 58.4%, rather than woody cover, which ranges more narrowly from 15% to 19.4%.

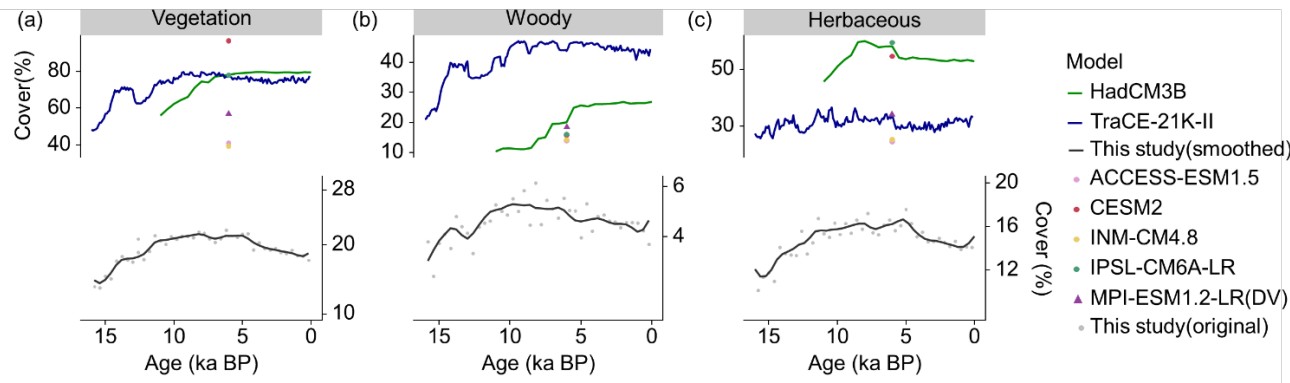

**Figure 6: Changes in vegetation cover of model-prescribed or simulations and reconstruction dataset since 16 Ka BP.** Total vegetation cover (a), Woody cover (b), Herbaceous cover (c). The gray lines indicate cover at a 400-year resolution. The blue lines

represent the smoothed changes in cover using the LOESS method. Circular indicates models using prescribed preindustrial vegetation cover, whereas triangular indicates models with activated dynamic vegetation modeling. Transition simulations (TraCE-21K-II and HadCM3B) both use dynamic vegetation.

## 4 Discussion

### 4.1 Reliability of machine learning-based reconstruction of vegetation cover

Here we employed five machine learning methods and the Modern Analogy Technique (all models used default parameters) for reconstructing temporal trends. Among these, RF models achieved the highest $R^2$ values and the lowest RMSE and MAE, followed by extreme gradient boosting, Modern Analogy Technique, k-nearest neighbor, gradient boosting machine, and support vector machines (Fig. S2), consistent with other studies (Hengl et al., 2018; Lindgren et al., 2021). This study suggested that the RF model is a superior method for reconstructing vegetation cover using pollen data.





We employed a comprehensive pollen dataset from both the Tibetan Plateau to develop an RF-temporal model at the site level. This extensive modern surface pollen database across Asia spans the spatial climate gradient that could be large enough to encompass the temporal one recorded by fossil pollen assemblages (Fig. S6), giving us a relatively high level of confidence in the reconstruction of vegetation cover at the site level. In addition, the inclusion of topographic variables in developing the RF-temporal model could significantly improve the predictive accuracy at the site level (Fig. S7).

By extrapolating vegetation cover from the site level to the spatial scale, we first develop an individual RF-spatial model for each 400-year time bin (Fig. S8) and used gridded climate and topographic data from paleoclimatic simulations to obtain spatially continuous vegetation cover within each time bin. We found that ESMs generally performed much better in capturing spatial variation in paleoclimatic variables than its temporal variability. The notorious model errors in temporal variability of paleoclimatic variables would not greatly affect our reconstruction within each time bin, since we only use the

spatial pattern on paleoclimate variables in spatial interpolation. This statement was further confirmed by our perturbation tests. Specifically, within each 400-year bin, we developed 20 sets of spatial RF models by using the fossil pollen data within this bin as the response variable and randomly selecting paleoclimate data from other bins as drivers. These perturbation results were generally consistent with the original results (Fig. S9), suggesting that the temporal variability in gridded climate data would not affect the temporal variability in our reconstruction.

However, our pollen-based reconstruction still suffered from certain uncertainties. First, our reconstruction of paleo-vegetation cover relies on the assumption that the relationship between pollen records and vegetation cover, extracted from modern observations, has remained consistent over time. This assumption implied that the pollen productivity estimates (PPEs) and the Relevant Source Area of Pollen (RSAP) for modern vegetation are similar to those for past vegetation (Sugita, 2007).

Second, the RF models tend to overestimate low values and underestimate high values (Wang et al., 2023; Liu et al., 2024). The spatially uneven distribution of surface pollen samples would exacerbate this problem. In addition, the long-distance transport of arboreal pollen from forested regions at lower altitudes may lead to an overestimate of vegetation cover in receptor regions (Wang et al., 2023). An alternative solution is to create mock records  (Hengl et al., 2018; Lindgren et al., 2021). For instance, fossil pollen records are inherently sparse in barren regions (e.g., alpine glaciers and the deserts of the

Tarim Basin), while we could assume that these areas have been devoid of vegetation during certain periods. Adding sample points across these unvegetated regions would enhance model performance in the prediction of vegetation cover. Moreover, incorporating records of past desert regions from other paleo-evidence beyond pollen could further improve accuracy (Davis et al., 2024).

        Third, surface pollen samples could potentially be corrupted by anthropogenic disturbances, such as land use, agricultural

practices, and the introduction of exotic plants (Cronin et al., 2017; Sobol et al., 2019a). In addition, for remote sensing data on vegetation cover, imaging issues in MODIS data can introduce significant uncertainties, particularly in estimating grass cover beneath the tree canopy (Liu et al., 2017).



## 4.2 Applications of spatio-temporally explicit estimate of vegetation cover

Here we reconstruct the first spatiotemporally continuous vegetation cover dataset using random forest. The
spatiotemporally continuous vegetation cover datasets provide a millennial-scale perspective on how vegetation responds
and adapts to paleoclimatic change on the one hand (Xu et al., 2023; Dziomber et al., 2024). On the other hand, by analyzing
the woody-to-herbaceous ratio in our reconstruction, we could potentially reveal how westerlies and Asian monsoons
evolved over the Tibetan Plateau since LGM (Sun et al., 2017). In addition, the vegetation cover presented in this dataset is a
result of impacts from both paleoclimatic change and prehistorical human activities. By comparing pollen-based
reconstruction to pure climate change-induced changes in vegetation cover (e.g., ESMs results), we could identify the onset
and magnitude of human activities on the Tibetan Plateau (Strandberg et al., 2023).

Second, the comparison of our reconstruction with vegetation cover in ESMs over the Tibetan Plateau shows that the models
generally overestimate vegetation-related variables, which is linked to inaccurate parameterization of soil moisture dynamics
(Yang et al., 2020; Song et al., 2021; Kang et al., 2022). Such overestimation would introduce a significant bias into
simulations of surface radiation balance, water, energy, and carbon cycles (Alibakhshi et al., 2020; Gui et al., 2024). For
instance, models generally overestimate vegetation cover in the western plateau, which suggests that models have a lower-
than-expected surface albedo and then a notable climate bias. Evidence is mounting that surface darkening over the Tibetan
plateau could enhance Asian monsoon systems (Tang et al., 2023b). The lower-than-expected albedo in models could then
introduce a bias into simulations of atmospheric circulation and precipitation patterns over Asian regions (Tang et al., 2023b).
Prescribing our spatio-temporally explicit map in ESMs could help realistically capture the biophysical and biogeochemical
impacts of vegetation cover changes on paleoclimatic change.

## 5 Data availability

Data are publicly accessible at the zenodo via the following link: https://doi.org/10.5281/zenodo.14211026 (zhang, 2024).
This link provides a detailed data summary along with instructions on variable definitions in file and their usage, ensuring
that readers can effectively utilize the dataset.

## 6 Conclusions

Here we integrate fossil pollen assemblages, along with the relationship between modern pollen records and vegetation cover,
in a machine-learning approach to generate a spatio-temporally explicit map of vegetation cover (total vegetation, woody
plants, and herbaceous plants) for the Tibetan Plateau, spanning from the deglaciation period to the present, at a spatial
resolution of 0.5° and a temporal resolution of 400 years. We discussed how different settings of random forest modeling
affect reconstruction accuracy, and demonstrated the robustness of our pollen-based reconstruction. In contrast to the
previous pollen-based reconstruction at the site level over the Tibetan Plateau, we have produced the most spatially complete



estimate by ingesting spatial information on climate variables. We demonstrated that the use of spatial information on paleoclimatic data in producing the temporal evolution of regional vegetation cover would not be affected by notorious
uncertainties in the temporal evolution of paleoclimatic variables. Our machine learning-based vegetation cover dataset can be used to understand how vegetation responds and adapts to paleoclimatic change. Moreover, this vegetation data can also be fed into the Earth system models for quantifying the "true" feedback of vegetation cover changes on paleoclimatic change.

**Financial support.**

This research has been supported by the Second Tibetan Plateau Scientific Expedition and Research Program
[2022QZKK0101; 2024QZKK0301] and the National Natural Science Foundation of China [42425106].

**Competing interests.**

The contact author has declared that neither they nor their co-authors have any competing interests.

**Author contribution**

PC conceived the idea for the study, and developed the concept together with TW. PC performed the data analysis and wrote
the manuscript, with major contributions provided by TW. All the authors contributed to the discussions and paper revision.



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
