# Peer review of "Pollen-based reconstruction of spatially-explicit vegetation cover over the Tibetan Plateau since the last deglaciation"

_Earth System Science Data, 2024_

## Author Comment (AC1)

**To Reviewer #2**

**General Comments**

**[Comment 1]** *This paper provides a spatio-temporally map of vegetation cover on the Tibetan Plateau. These data would be very useful for the Holocene terrestrial ecosystem studies of the Plateau.*

**[Response]** Many thanks for your encouragement, constructive comments, and suggestions. The point-by-point responses are listed following each comment.

**[Comment 2]** *In Modern pollen dataset, some of the samples from same paper are given the same coordinates, because in some old references the authors didn't provide precise locations of each sample. But if these samples were from mountainous area, the vegetation at each site are certainly different. How do the authors deal with these sites, will it affect the vegetation reconstruction?*

**[Response]** Thank you for your insightful comment. The modern pollen dataset used in this study was compiled by Cao et al. (2022) from multiple regional modern pollen datasets, which underwent rigorous quality control during collection and integration. Low-quality samples (e.g., extremely low pollen concentration, very few taxa, incomplete data, or pollen percentage sums outside 90–110%) were removed, and the datasets were standardized by calibrating geographic coordinates and harmonizing taxon names. The majority (>90%) of samples were collected after the 1990s using GPS, ensuring precise coordinates. For older samples, geographic positions were estimated using topographic maps or proportional maps and further corrected using

DEM-based elevation fitting and Google Earth satellite imagery (Zhuo et al., 2013). These datasets have been widely used for paleoecological reconstructions and demonstrated high accuracy, supporting their reliability (Yu et al., 2000; Cao et al., 2013, 2014; Chen et al., 2021; Liu et al., 2020; Wang et al., 2022).

Considering the possibility of duplicate records across datasets and the fact that modern pollen samples from lake sediment surfaces may originate from multiple samples of a single fossil pollen record after 1950 CE, we averaged the pollen percentages of samples sharing identical coordinates but containing different taxa. Following your suggestion, we then performed a sensitivity analysis to assess whether retaining these samples would bias vegetation reconstruction. Specifically, we identified records with duplicate coordinates (n = 1011 samples, representing 330 coordinate sites or 4.3% of all sites), including 133 sites (1.8%) located in mountainous areas as defined by the global mountain of Kapos et al. (2000), which objectively accounts for altitude, slope, and relative relief (Sayre et al., 2018).

We constructed two Random Forest (RF) models based on (i) all sites (n = 7587) and (ii) sites with duplicate coordinates removed (n = 7287. The results showed that removing duplicate sites did not affect model performance: the accuracy of RF models was the same ($R^2$ = 0.79), and reconstructed vegetation cover was highly consistent ($R^2$ = 0.99) between the two models (Figure R1). Therefore, the inclusion of a small proportion of modern pollen sites with identical coordinates does not compromise the accuracy of vegetation reconstruction. This information has been incorporated into the revised manuscript.

[Figure]

**Figure R1.** Model validation and vegetation cover reconstruction with and without duplicate-coordinate modern pollen sites. (**a-b**) Accuracy of RF models trained using all sites (a) versus those with duplicates removed (b). (**c**) Comparison of reconstructed vegetation cover between models with and without duplicate-coordinate sites.

**[Comment 3]** *Moreover, some of the sites' coordinates are very rough, e.g. only with degree, would these represent the real localities and true vegetation type? did the author deleted these?*

**[Response]** Thanks for your thoughtful comment. To ensure accurate correspondence between modern pollen sites and true vegetation cover, we have removed sites with coordinates reported only at the degree level (n = 245, accounting for 3.1% of all sites).

To further verify the robustness of using the remaining modern pollen sites as training data for vegetation reconstruction models, we conducted a sensitivity test by introducing random perturbations to site coordinates. Specifically, we randomly selected 10% of the pollen sites—only those from the 1990s dataset exhibited potential coordinate uncertainty, and these accounted for a very small proportion of the dataset—and applied random shifts within a range of 0 to 0.05° (~5.6 km). The

choice of 0.05° is reasonable, as coordinates derived from map-based estimates

typically have an error of about ±2 km depending on latitude (Whitmore et al., 2005).

We then re-extracted vegetation cover values using the perturbed coordinates,

reconstructed Random Forest (RF) models, and compared the results to models based

on the original coordinates.

Our perturbation tests showed that the RF models built with perturbed

coordinates achieved comparable accuracy to those based on the original dataset ($R^2 =$

0.79), and the reconstructed vegetation cover exhibited highly consistent ($R^2 = 1$)

(Figure R2). This consistency likely results from the strong spatial autocorrelation in

modern vegetation cover datasets derived from remote sensing products, where

adjacent pixels exhibit similar vegetation composition. Additionally, vegetation

distribution patterns at regional scales typically form continuous belts or patches

(Turner et al., 2001). Therefore, even when pollen site coordinates have slight

uncertainties, the associated vegetation coverage values remain relatively stable,

confirming the robustness of our modern pollen dataset for RF-based vegetation

reconstruction. This information has been added to the revised manuscript.

[Figure]

**Figure R2.** Model validation and vegetation cover reconstruction based on modern

pollen sites with randomly perturbed coordinates. (**a-b**) Accuracy of RF models

trained using original (a) and perturbed (b) coordinates. (**c**) Comparison of reconstructed vegetation cover between models using original and perturbed coordinates.

**Specific Comments**

[**Comment 4**] *There is still few Chinese characters in the text, i.e. line 112.*

[**Response**] Thank you for pointing this out. We have carefully checked the manuscript and replaced the remaining Chinese character with its correct English equivalent (Line 134, Page 5).

**Reference**

Cao, X., Ni, J., Herzschuh, U., Wang, Y., and Zhao, Y.: A late Quaternary pollen dataset from eastern continental Asia for vegetation and climate reconstructions: Set up and evaluation, Rev. Palaeobot. Palynol., 194, 21–37, https://doi.org/10.1016/j.revpalbo.2013.02.003, 2013.

Cao, X., Herzschuh, U., Telford, R. J., and Ni, J.: A modern pollen–climate dataset from China and Mongolia: Assessing its potential for climate reconstruction, Rev. Palaeobot. Palynol., 211, 87–96, https://doi.org/10.1016/j.revpalbo.2014.08.007, 2014.

Cao, X., Tian, F., Herzschuh, U., Ni, J., Xu, Q., Li, W., Zhang, Y., Luo, M., and Chen, F.: Human activities have reduced plant diversity in eastern China over the last two millennia, Glob. Change Biol., 28, 4962–4976, https://doi.org/10.1111/gcb.16274, 2022.

Chen H.-Y., Xu D.-Y., Liao M.-N., Li K., Ni J., Cao X.-Y., Cheng B., Hao X.-D., Kong Z.-C., Li S.-F., Li X.-Q., Liu G.-X., Liu P.-M., Liu X.-Q., Sun X.-J., Tang L.-Y., Wei H.-C., Xu Q.-H., Yan S., Yang X.-D., Yang Z.-J., Yu G., Zhang Y., Zhang Z.-Y., Zhao K.-L., Zheng Z., and Ulrike H.: A modern pollen dataset of China, Chin. J. Plant Ecol., 45, 799–808, https://doi.org/10.17521/cjpe.2021.0024, 2021.

Kapos, V., Rhind, J., Edwards, M., Price, M. F., and Ravilious, C.: Developing a map of the world's mountain forests., in: Forests in sustainable mountain development: a state of knowledge report for 2000. Task Force on Forests in Sustainable Mountain Development., 4–19, https://doi.org/10.1079/9780851994468.0004, 2000.

Liu, L., Wang, W., Chen, D., Niu, Z., Wang, Y., Cao, X., and Ma, Y.: Soil-surface pollen assemblages and quantitative relationships with vegetation and climate from the Inner Mongolian Plateau and adjacent mountain areas of northern China, Palaeogeogr. Palaeoclimatol. Palaeoecol., 543, 109600, https://doi.org/10.1016/j.palaeo.2020.109600, 2020.

Sayre, R., Frye, C., Karagulle, D., Krauer, J., Breyer, S., Aniello, P., Wright, D. J., Payne, D., Adler, C., Warner, H., VanSistine, D. P., and Cress, J.: A New High-Resolution Map of World Mountains and an Online Tool for Visualizing and Comparing Characterizations of Global Mountain Distributions, Mt. Res. Dev., 38, 240–249, https://doi.org/10.1659/MRD-JOURNAL-D-17-00107.1, 2018.

Turner, M. G., Gardner, R. H., and O'Neill, R. V. (Eds.): Organisms and Landscape Pattern, in: Landscape Ecology in Theory and Practice: Pattern and Process, Springer, New York, NY, 201–247, https://doi.org/10.1007/0-387-21694-4_8, 2001.

Wang, N., Liu, L., Zhang, Y., and Cao, X.: A modern pollen data set for the forest–meadow–steppe ecotone from the Tibetan Plateau and its potential use in past vegetation reconstruction, Boreas, 51, 847–858, https://doi.org/10.1111/bor.12589, 2022.

Whitmore, J., Gajewski, K., Sawada, M., Williams, J. W., Shuman, B., Bartlein, P. J., Minckley, T., Viau, A. E., Webb, T., Shafer, S., Anderson, P., and Brubaker, L.: Modern pollen

data from North America and Greenland for multi-scale paleoenvironmental applications, Quat. Sci. Rev., 24, 1828–1848, https://doi.org/10.1016/j.quascirev.2005.03.005, 2005.

Yu, G., Chen, X., Ni, J., Cheddadi, R., Guiot, J., Han, H., Harrison, S. P., Huang, C., Ke, M., Kong, Z., Li, S., Li, W., Liew, P., Liu, G., Liu, J., Liu, Q., Liu, K.-B., Prentice, I. C., Qui, W., Ren, G., Song, C., Sugita, S., Sun, X., Tang, L., Van Campo, E., Xia, Y., Xu, Q., Yan, S., Yang, X., Zhao, J., and Zheng, Z.: Palaeovegetation of China: a pollen data-based synthesis for the mid-Holocene and last glacial maximum, J. Biogeogr., 27, 635–664, https://doi.org/10.1046/j.1365-2699.2000.00431.x, 2000.

Zhuo Z., Kangyou H., Jinhui W., Yuanfu Y., Qiuchi W., Qinghai X., Houyuan L., Yunli L., Chuanxiu L., Yanwei Z., Chunhai L., Shixiong Y., Jie L., Anding P., Yun D., Haicheng W., Beaudouin C., Tarasov P., Nakagawa T., and Cheddadi R.: MODERN POLLEN DATA IN CHINA AND ADJACENT AREAS:SPTATIAL DISTRIBUTION FEATURES AND APPLICATIONS ON QUANTITATIVE PALEOENVIRONMENT RECONSTRUCTION, Quat. Sci., 33, 1037–1053, https://doi.org/10.3969/j.issn.1001-7410.2013.06.01, 2013.

---

## Author Comment (AC2)

**To Reviewer #1**

**General Comments**

**[Comment 1]** *The manuscript presented a spatiotemporally contiguous palaeovegetation cover dataset of the Tibetan Plateau since the 16ka BP, generated by the temporal and temporal random forest (RF) models using modern pollen and fossil pollen records. This is the first mapping of past vegetation on the Tibetan Plateau which should be benefit to the palae-community. I believe that the methods used in this study is feasible and the results are robust.*

**[Response]** Many thanks for your careful review and constructive suggestions. We greatly appreciate your insightful comments, which have significantly improved the manuscript. We hope our revisions and explanations have satisfactorily addressed your questions and comments.

**[Comment 2]** *However, given the complexity of the vegetation of the TP, this study only reconstructed two types of vegetation, woody and herb vegetation, plus the total vegetation, which were too coarse for further use. The pollen records can be transformed to multiple vegetation types on the plateau, at least for example, forest, coniferous forest, shrubland and tundra or various alpine vegetation (alpine meadow, steppe and desert). The reviewer suggests that the authors should reconsider the classification of vegetation types as fine as possible.*

**[Response]** Thank you for this insightful suggestion. In response, we refined our vegetation classification scheme better to reflect the complexity of vegetation on the

Tibetan Plateau. Specifically, we disaggregated the original "woody" category into coniferous forest and broadleaved forest, and the "herbaceous" category into alpine meadow and alpine steppe. In total, we reconstructed seven vegetation types: total vegetation, woody plants, herbaceous plants, broadleaved forest, coniferous forest, alpine meadow, and alpine steppe.

To estimate modern vegetation cover for these types, we used the Global Land Surface Satellite (GLASS) product, extracting total vegetation cover within circular buffers around modern pollen sites (5 km radius for surface soil samples, 50 km for lake sediments). We derived fractional coverage of broadleaved forest, coniferous forest, and herbaceous plants from the MODIS Land Cover Type Product (MCD12Q1), which provides an annual plant functional type (PFT) classification. Specifically, we grouped deciduous broadleaved and evergreen broadleaved forests as "broadleaved forest", and grouped deciduous coniferous and evergreen coniferous forests as "coniferous forest," due to their relatively small spatial extents on the Tibetan Plateau (<7% and <1%, respectively; Figure R1). Furthermore, shrubs and trees were grouped as "woody", rather than being shrubs as a separate vegetation type, as their distribution across the Tibetan Plateau accounts for less than 1%.

We calculated the proportion of each vegetation type within pollen-site buffers and multiplied these by the GLASS total vegetation cover to estimate the fractional cover of broadleaved forest, coniferous forest, and herbaceous plants. To further distinguish alpine meadow and alpine steppe—two ecologically important grassland types in the region—we used the updated Vegetation Map of China (1:1000000) (Su

et al., 2020) to determine their relative proportions within herbaceous areas, and then apportioned herbaceous cover accordingly.

Corresponding updates have been made in both the Methods and Results sections of the revised manuscript.

[Figure]

**Figure R1**. Spatial distribution of modern vegetation cover for different plant functional types (PFTs) on the Tibetan Plateau.

**Specific Comments**

**[Comment 3]** *Line 106-108, we used the MODIS Land Cover Type Product (MCD12Q1), which provides an annual Plant Functional Type (PFT) classification (DiMiceli et al., 2022). Trees were classified as "woody", while shrubs and grasses were grouped as "herbaceous." Why not keep the tree, shrub and grass PFTs? These three PFTs should be the appropriate vegetation types of the Tibetan Plateau. Otherwise, trees and shrubs should be classified as "woody", while grasses were grouped as "herbaceous."*

**[Response]** Thank you for your comment. In our revised analysis, we redefined

"woody" as including both trees and shrubs, and "herbaceous" as including only grasses, following the reviewer's recommendation.

Regarding the classification of shrub as a separate PFT, the reason it was not initially treated independently is that the product of MODIS (MCD12Q1) and GLASS data resulted in a very low average shrub cover (<1%) across the Tibetan Plateau (Figure R1), making it difficult to reconstruct reliable spatial and temporal patterns for this category alone. We have refined our classification scheme and now reconstruct seven vegetation types separately: total vegetation, woody plants, herbaceous plants, broadleaved forest, coniferous forest, alpine meadow, and alpine steppe (see detailed responses to **Comment 2** by ***Reviewer #1***). These categories better reflect the ecological diversity of the Tibetan Plateau and are supported by both pollen data and remote sensing products.

**[Comment 4]** *Line 117-118, for the palaeovegetation models, we standardized all forest vegetation types in the models as woody and all grass and shrub vegetation types as herbaceous. This is not acceptable too.*

**[Response]** Thank you for your comment. To ensure consistency between pollen-based reconstructions and model-simulated paleo-vegetation cover, we revised the classification scheme used in the vegetation models. Specifically, trees and shrubs were grouped as "woody", and grasses were classified as "herbaceous", consistent with common plant functional type (PFT) conventions.

Unlike the pollen-based reconstructions, which included seven PFTs (e.g., total

vegetation, woody plants, herbaceous plants, broadleaved forest, coniferous forest, alpine meadow, and alpine steppe), the vegetation outputs from the vegetation models did not include alpine-specific types such as steppe and meadow. As such, only the five shared PFT categories were used for comparison between reconstructed and simulated vegetation cover.

Corresponding changes have been made to both the Methods and Results sections of the revised manuscript to reflect this harmonized classification approach.

[Comment 5] *How to select the past vegetation type from a fossil pollen record at 400-year interval?*

[Response] Thank you for your constructive comment. To account for vegetation adaptation to climate changes over the past 20,000 years, we reconstructed palaeovegetation cover for each 400-year interval by building separate RF models to predict spatial vegetation patterns for that time slice. Therefore, the quality of fossil pollen records in each time interval is critical for model reliability.

We first evaluated the temporal resolution of 61 filtered fossil pollen sequences, which showed a median resolution of 220 years and a 75th percentile of 360 years (Figure R2). We selected a 400-year time interval as a compromise between two key considerations: (i) avoiding overly fine intervals that would require interpolation in RF-temporal reconstructions and introduce additional uncertainties into RF-spatial models, and (ii) ensuring sufficient sample size for each time bin. Our analysis revealed that fossil pollen sample counts increase almost linearly with coarser bin

widths between 100 and 400 years, but the gain becomes marginal beyond 400 years; for 400-600 year intervals, the sample sizes per bin are similar (Figure R3).

To further assess the robustness of this choice, we compared RF-spatial model performance at resolutions of 200, 300, 400, 500, and 600 years (Figure R4). The results showed that 400-year bins yielded predictive accuracy comparable to 500–600-year bins but better captured vegetation responses to centennial-scale climatic events, such as the Younger Dryas and Bølling–Allerød periods (Figure R5).

In summary, the 400-year interval represents an optimal balance between model accuracy and the temporal resolution needed to resolve rapid vegetation changes. Nevertheless, future research would greatly benefit from more in *situ* fossil pollen records with higher temporal resolution, as current datasets remain limited in both time and space. All of this information has been added to the revised manuscript.

[Figure]

**Figure R2**. The statistical distribution of the resolution and temporal coverage of the fossil pollen records. The horizontal black line indicates the mean values, bar ends represent the 25th and 75th percentiles, and horizontal lines represent the 5th and 95th percentiles.

[Figure]

**Figure R3.** Changes in the number of fossil pollen records from 16 ka to the present at different temporal resolutions.

[Figure]

**Figure R4.** Model performance of Random Forest reconstructions across different

temporal resolutions based on 10-fold cross-validation.

[Figure]

**Figure R5.** Temporal trends of reconstructed vegetation cover under different

temporal resolutions. Solid lines represent regional means estimated using a

Generalized Additive Model, and shaded areas indicate 95% confidence intervals.

**[Comment 6]** *For the RF-temporal models, why have only pollen and topographic*

*factors been used? Why not add climate data? But for the RF-spatial models, all the*

*data f climate and topography were applied.*

**[Response]** Thank you for this important comment. The RF-temporal models were designed to reconstruct temporal changes in vegetation cover at individual site scales. Importantly, this approach is independent of climatic inputs or climate model assumptions, ensuring that our palaeo-reconstructions directly reflect past floristic changes preserved in the sedimentary pollen record. Climate simulations from Earth system models often exhibit biases in temporal trends when compared with proxy-based reconstructions, such as the well-known Holocene temperature conundrum(Liu et al., 2014; Kaufman & Broadman, 2023). Incorporating such climate data into RF-temporal models could therefore distort the true temporal patterns of vegetation change inferred from pollen.

In contrast, climate data are essential for RF-spatial models, which aim to reconstruct the spatial distribution of vegetation at specific time slices. Climate variables provide continuous spatial fields that help capture the regional-scale environmental gradients influencing vegetation patterns. Furthermore, our sensitivity analysis confirmed that the use of gridded climate data in RF-spatial models does not significantly affect the temporal variability of our spatio-temporal dataset. Specifically, we developed 20 sets of RF-spatial models using palaeovegetation cover as the response variable while randomly assigning palaeoclimate data from non-corresponding time bins as predictors. The reconstructed vegetation cover was generally consistent with that from the original models (Figure R6).

[Figure]

**Figure R6.** The comparison between the perturbation experiment and the normal

experiment. In the perturbation experiment, the temporal sequence of input data used

for RF-spatial is randomly scrambled.

**[Comment 7]** *Line 298, zhang 2024 to Zhang 2024.*

**[Response]** Thank you for pointing this out. We have corrected "zhang 2024" to

"Zhang 2024" in the revised manuscript.

**Reference**

Kaufman, D. S., & Broadman, E. (2023). Revisiting the Holocene global temperature conundrum. *Nature*, *614*(7948), 425–435. https://doi.org/10.1038/s41586-022-05536-w

Liu, Z., Zhu, J., Rosenthal, Y., Zhang, X., Otto-Bliesner, B. L., Timmermann, A., Smith, R. S., Lohmann, G., Zheng, W., & Elison Timm, O. (2014). The Holocene temperature conundrum. *Proceedings of the National Academy of Sciences*, *111*(34), E3501–E3505. https://doi.org/10.1073/pnas.1407229111

Su, Y., Guo, Q., Hu, T., Guan, H., Jin, S., An, S., Chen, X., Guo, K., Hao, Z., Hu, Y., Huang, Y., Jiang, M., Li, J., Li, Z., Li, X., Li, X., Liang, C., Liu, R., Liu, Q., … Ma, K. (2020). An updated Vegetation Map of China (1:1000000). *Science Bulletin*, *65*(13), 1125–1136. https://doi.org/10.1016/j.scib.2020.04.004

---

## Author Response (AR2)

**To Reviewer #1**

**General Comments**

[Comment 1] Authors have improved the manuscript according to comments from reviewers. I basically satisfy most of the modifications, especially the updated vegetation classification scheme. However, such classification still has some shortfalls, such as the study couldn't separate evergreen and deciduous forests from broadleaved and coniferous forests, and couldn't separate alpine shrubland and desert from alpine meadow and steppe. The reviewer understands that from the global remote sensing datasets and also from pollen data, such separations are very hard. So, the authors should discussion how such shortfalls of this vegetation scheme could affect the interpolation of the detailed vegetation change on the TP.

[Response] Thank you for your constructive comment and understanding. As the reviewer noted, the updated vegetation classification scheme represents a compromise between broader applicability and accuacy, constrained by the limitations of both modern global remote-sensing datasets and pollen datasets. In the revised manuscript, we have explicitly discussed these limitations and their implications for the application of our dataset. Specifically, we added the following text: "The accuracy of both pollen- and remote-sensing-based vegetation classifications imposes constraints on the vegetation classification scheme of our reconstruction. Pollen identification, relying primarily on exine morphology, is typically limited to the genus or family level, making it difficult to distinguish functional ecological traits such as evergreen versus deciduous. For example, evergreen and deciduous species of *Quercus* display

only minor morphological differences in their pollen and are therefore generally grouped as "Quercus-type pollen" (Peñuelas et al., 2009), with finer distinctions requiring additional evidence such as macrofossils (Liu et al., 2007). This limitation is compounded by the relatively coarse spatial resolution of MODIS data, while suitable for regions with simple land cover types, often yields low classification accuracy in areas with more complex vegetation (Zeng et al., 2016). As a result, we did not separate evergreen and deciduous forests from broadleaved and coniferous forests. Additionally, all existing global land cover datasets consistently indicate that shrub cover on the Tibetan Plateau is minimal, generally less than 2% (Yang et al., 2017). Accordingly, shrubs were not treated as a separate class but instead merged with trees under the broader category of woody vegetation. The inability to separate evergreen and deciduous forests constrains applications in climate, carbon, and biodiversity studies, as well as paleoecological reconstructions of vegetation responses to environmental disturbances." (Lines 324-336 on Pages 15-16)

Furthermore, we emphasized that coupling data-driven and process-based approaches provides a promising direction for future improvement. For instance, process-based models such as REVEALS can generate taxon-specific cover estimates by accounting for pollen productivity and dispersal, whereas data-driven approaches provide actual vegetation cover to calibrate and refine these reconstructions. Such integration not only enhances the robustness and reliability of the dataset through cross-validation of independent methods, but also allows taxon-level reconstructions that enable tracing of species' migrations, expansions, and contractions in response to

climatic transitions. We consider this integration an important avenue for our future work. All of this information has been clarified in the revised manuscript.

**Specific Comments**

[Comment 2] Another suggestion: woody plants can be changed to woody vegetation, and herb plants to herb vegetation. These new terms match with the terms of other forest and grassland vegetation.

[Response] Thank you for your helpful suggestion. In the revised manuscript, we have adopted the terms "woody vegetation" and "herb vegetation" in place of "woody plants" and "herb plants," respectively.

**To Reviewer #2**

**General Comments**

[Comment 1] I concur with the previous reviewers' observations. While the dataset provides valuable information, there remains substantial uncertainty associated with the original sample data, the reconstruction procedures, and the interpolation, as well as the temporal and spatial extrapolations. A more comprehensive discussion of the data sources and methodologies is warranted, together with explicit cautionary notes on the appropriate use and interpretation of these data.

[Response] Thank you for your thoughtful suggestion. Following your advice, we have added a new subsection in the revised manuscript to systematically address the uncertainties in paleovegetation reconstructions, including those arising from pollen datasets, remote-sensing-derived vegetation cover, land-cover products, and methodological limitations, as the following text: "Although data-driven machine learning methods provide a less parameter-intensive approach to reconstructing paleovegetation, they still rely on the assumption that the relationship between pollen records and vegetation cover, extracted from modern observations, has remained consistent over time. Consequently, the robustness of our reconstruction ultimately depends on the quality of the input datasets, including pollen percentage data and vegetation cover derived from remote sensing observations.

Pollen datasets, compiled from diverse studies with varying objectives and methods, inevitably contain inconsistencies in sampling, taxonomic identification, and age control. We implemented rigorous quality-control procedures, including duplicate

removal, correction of inaccurate coordinates, taxonomic standardization, and filtering for higher temporal resolution and reliable chronologies. Nonetheless, unavoidable uncertainties remain due to environmental contamination and the absence of standardized pollen processing and identification protocols. Furthermore, pollen samples could potentially be corrupted by anthropogenic disturbances, such as land use, agricultural practices, and the introduction of exotic plants (Cronin et al., 2017; Sobol et al., 2019a; Zhang et al., 2025).

To link modern pollen assemblages with vegetation cover, we employed the MCD12Q1 land cover product in conjunction with the GLASS vegetation cover dataset to estimate the cover of different plant functional types. While these satellite-based reanalysis datasets are robust at the global scale and are widely applied, they generally introduce larger uncertainties than field observations or region-specific vegetation maps. Such acceptable but non-negligible errors inevitably affect the precision of paleovegetation reconstructions. In addition, although the majority of modern pollen samples used in this study were collected after the 2000s, some were obtained in the 1980s and 1990s. The vegetation represented by these earlier samples may have shifted under contemporary climate change, particularly given the rapid warming observed in recent decades.

The accuracy of both pollen- and remote-sensing-based vegetation classifications imposes constraints on the vegetation classification scheme of our reconstruction. Pollen identification, relying primarily on exine morphology, is typically limited to the genus or family level, making it difficult to distinguish

functional ecological traits such as evergreen versus deciduous. For example, evergreen and deciduous species of Quercus display only minor morphological differences in their pollen and are therefore generally grouped as "Quercus-type pollen" (Peñuelas et al., 2009), with finer distinctions requiring additional evidence such as macrofossils (Liu et al., 2007). This limitation is compounded by the relatively coarse spatial resolution of MODIS data, while suitable for regions with simple land cover types, often yields low classification accuracy in areas with more complex vegetation (Zeng et al., 2016). As a result, we did not separate evergreen and deciduous forests from broadleaved and coniferous forests. Additionally, all existing global land cover datasets consistently indicate that shrub cover on the Tibetan Plateau is minimal, generally less than 2% (Yang et al., 2017). Accordingly, shrubs were not treated as a separate class but instead merged with trees under the broader category of woody vegetation. The absence of a more detailed vegetation classification scheme constrains the applications of this paleovegetation dataset in climate, carbon cycle, and biodiversity studies, particularly when differences among vegetation types are of primary interest." (Lines 304-336 on Pages 15-16)

In addition, in light of the uncertainties discussed above, we have summarized explicit cautionary notes on the appropriate use and interpretation of the dataset. The following text has been added to the revised manuscript: "The construction methods and spatiotemporal resolution of this dataset necessitate several considerations in its application: (1) The magnitude of modern vegetation cover datasets directly influences the magnitude of reconstructed vegetation cover, while their spatial

heterogeneity shapes the temporal variability of reconstructed sequences. However, substantial discrepancies exist among vegetation cover datasets owing to differences in data sources, processing methods, and classification systems (Liu et al., 2024b; Xu et al., 2024). Therefore, when comparing paleovegetation reconstructions derived from different modern vegetation cover datasets, these intrinsic differences must be carefully taken into account. (2) The spatiotemporal resolutions of  $0.5^{\circ} \times 0.5^{\circ}$  and 400 years are appropriate for regional- to continental-scale analyses and for examining long-term trends, but they are insufficient to resolve fine-scale ecological heterogeneity or capture decadal climatic fluctuations. (3) The classification of vegetation into seven types, while facilitating comparability with Earth System Models, does not fully capture the complexity of plant functional diversity or ensure direct equivalence with model-specific plant functional types." (Lines 479-389 on Page 17).